# Sociodemographic and Illness-Related Indicators to Predict the Status of Neuromyelitis Optica Spectrum Disorder (NMOSD) Five Years after Disease Onset

**DOI:** 10.3390/jcm11030734

**Published:** 2022-01-29

**Authors:** Dena Sadeghi-Bahmani, Mahdi Barzegar, Omid Mirmosayyeb, Saeed Vaheb, Nasim Nehzat, Vahid Shaygannejad, Serge Brand

**Affiliations:** 1Department of Psychology, University of Stanford, Stanford, CA 94305, USA; bahmanid@stanford.edu; 2Psychiatric Clinics, Center for Affective, Stress and Sleep Disorders, University of Basel, 4002 Basel, Switzerland; serge.brand@upk.ch; 3Sleep Disorders Research Center, Kermanshah University of Medical Sciences, Kermanshah 6714869914, Iran; 4Department of Neurology, School of Medicine, Isfahan University of Medical Sciences, Isfahan 8174673461, Iran; barzegar_mahdi73@yahoo.com (M.B.); omid.mirmosayyeb@gmail.com (O.M.); 5Isfahan Neurosciences Research Center, Isfahan University of Medical Sciences, Isfahan 8174673461, Iran; saeedvaheb.sv@gmail.com (S.V.); n.nehzat96@gmail.com (N.N.); 6Substance Abuse Prevention Research Center, Kermanshah University of Medical Sciences, Kermanshah 6714869914, Iran; 7Department of Sport, Exercise and Health, Division of Sport Sciences and Psychosocial Health, University of Basel, 4052 Basel, Switzerland; 8School of Medicine, Tehran University of Medical Sciences, Tehran 1417466191, Iran

**Keywords:** neuromyelitis optica spectrum disorder, predictors, long-term course, EDSS

## Abstract

Background: Neuromyelitis Optica Spectrum Disorder (NMOSD) is an autoimmune demyelinating disease of the central nervous system. Currently, no factors have been identified to predict the long-term course of NMOSD. To counter this, we analyzed data of 58 individuals with NMOSD at disease onset and about five years later. Methods: Medical records of 58 individuals with NMOSD (mean age: 31.13 years at disease onset; 86.2% female) were retrospectively analyzed. At baseline, a thorough medical and disease-related examination was performed; the same examination was repeated about five years later at follow-up, including treatment-related information. Mean outcome measure was the difference in EDSS (Expanded Disease Severity Scale) scores between baseline and follow-up. Results: Mean disease duration was 4.67 years. Based on the differences of the EDSS scores between baseline and follow-up, participants were categorized as improving (*n* = 39; 67.2%), unchanged (*n* = 13; 22.4%) and deteriorating (*n* = 6; 10.3%). Deteriorating was related to a higher progression index, and a higher number of attacks, while the annualized relapse rate reflecting the number of attacks per time lapse did not differ between the three groups. Improving was related to a higher intake of rituximab, and to a higher rate of seropositive cases. Unchanged was related to a lower rate of seropositive cases. Factors such as age, gender, somatic and psychiatric comorbidities, symptoms at disease onset, relapse rates, number and location of cervical plaques, or brain plaques and thoracolumbar plaques at baseline did not differ between those improving, deteriorating or remaining unchanged. Conclusions: Among a smaller sample of individuals with NMOSD followed-up about five years later, individuals deteriorating over time reported a higher progression index, while the annualized relapse rate was unrelated to the progress of disease. Overall, it appears that the course of NMOSD over a time lapse of about five years after disease onset is highly individualized. Accordingly, treatment regimen demands a highly individually tailored approach.

## 1. Introduction

Neuromyelitis optica (NMO) or Neuromyelitis Optica Spectrum Disorder (NMOSD) is an autoimmune inflammatory disorder, which prevalently affects the optic nerve and the spinal cord [1,2,3,4,5,6,7]. Accordingly, simultaneous or consecutive attacks of acute optic neuritis and transverse myelitis characterize NMOSD [7]. Individuals with NMOSD suffer from visual impairment/decreased visual acuity, spinal cord dysfunction (muscle weakness, reduced sensation, loss of bladder and bowel control), along with an acute and severe spastic weakness of the legs (paraparesis) or all four limbs (quadriparesis) [8,9]. Further, NMOSD belongs to a cluster of neurological disturbances such as multiple sclerosis (MS), myelin glycoprotein antibody disorders (MOGAD), and chronic recurrent idiopathic neuritis (CRION). Table 1 reports the main differences between NMOSD, MS, and MOGAD.

The detection of pathogenic antibodies to aquaporin-4, a water channel present on astrocytic foot processes mainly expressed in the optic nerves, brainstem, and spinal cord, helped to distinguish NMOSD from multiple sclerosis (MS) [5,10,11]. As regards the etiology, in more than 80% of cases pathogenic IgG autoantibodies to aquaporin 4 (AQP4-IgG) causes NMOSD [7]. Further, individuals with NMOSD often experience a relapsing course, and severe symptoms, which may lead to a severe disability, along with a decreased quality of life [12,13,14]. In a related vein, compared to healthy controls, individuals with NMOSD reported a lower quality of life, higher symptoms of anxiety and depression, and more symptoms of restless legs syndrome. Further, a longer illness duration and higher fatigue scores predicted poor sleep quality [15].

Predicting the course of a disease is crucial for the swift, precise and individually-tailored treatment. Here, we summarize findings to possibly predict the illness course of NMOSD.

For the outcome of plasma exchange to treat NMOSD, Aungsumart and Apiwattanakul [16] identified a minimal disability at baseline such as preserved reflexes and a short disease duration, and early treatment initiation as favorable predictors of a good clinical outcome. The early initiation of plasma exchange shortly after diseases onset was the strongest predictor of the outcome of severe attacks among 60 patients with NMOSD [17].

Among 21 individuals with NMOSD and longitudinally extensive transverse myelitis (LETM), no clinical predictors could be identified, suggesting that the combination of NMOSD and LETM at disease onset appeared to be a highly heterogeneous disease [18].

Thirteen patients with NMOSD and LETM were followed-up about four years later [19]. With appropriate medication treatment, the outcome after LETM attacks was satisfactory in 10 out of 13 patients; improvements were above all observed upon motor deficit on the EDSS scores, while the improvements of visual function were modest. Further, while the cerebrospinal fluid (CSF) analysis showed no irregularities in 8 out of 13 cases, and while the spinal MRI showed evidence of LETM in all patients, in contrast, brain MRI showed no irregularities in 7 out of 13 patients. The overall conclusion was that spinal MRI was essential for NMOSD diagnosis in the presence of LETM, and the immunosuppressive therapy reduced the frequency of attacks.

While the measurements of optic chiasma appeared a useful marker to estimate the degeneration of the anterior optic pathway among 39 patients with NMOSD, on the flip side, such measurements were not useful so far to predict the long-term course of NMOSD [20].

Royston et al. [21] surveyed 1363 patients with NMOSD (mean age: 44.9 years; 75.3% female) about two years after the first assessment. Of those, 47.7% had one or more relapses, with an annualized relapse rate (ARR) of 0.8. Royston et al. [21] also estimated the care costs and reported that the annual care costs per person was about USD 60,599, with an estimated cost of USD 10,070 per relapse. 

Compared to Royston et al. [21], Stellmann et al. [22] observed an annualized relapse rate (ARR) of 0.6 1.2 years after disease onset among 144 patients with NMOSD (mean age: 40.9 years; 82.6% female). Older age was associated with a lower risk of further attacks, while a previous attack under the same treatment medications predicted higher odds for a shorter inter-attack interval.

Kleiter et al. [23] investigated possible clinical patterns among 185 patients with NMOSD (82% females) reporting 871 attacks; the authors showed that particularly patients with myelitis and bilateral optic neuritis had lower remission rates, while the augmentation of medications favorably impacted on lower attack rates. More specifically as regards the medication treatment, eculizumab was superior to placebo to reduce the relapse rates, while eculizumab did neither accelerate nor slow down the disability progression [24].

To summarize, while meanwhile the etiopathology and pathophysiology of NMOSD appeared to be well understood [7], in contrast, the current literature showed that the identification of clinically useful biomarkers and further disease-related indices for the prediction of relapses and the therapeutic response appeared modest and unsatisfactory [6,7]. Further, the clinical course of the NMOSD causes a substantial burden for the patient, and such burdens are associated with substantial healthcare utilizations in addition to expenditure for the public sector [21]. Given this background, the aim of the present retrospective and observational cohort study was to identify possible sociodemographic and disease-related indices to predict the course of NMOSD among a sample of 58 patients with NMOSD over a time lapse of about five years. To more objectively operationalize the disease progress, the decision was to focus on the Expanded Disease Status Scale (EDSS) scores [25,26] at follow-up and to compare these scores with those at baseline, and this for the following reasons: First, neurologists experienced in the field of NMOSD and MS are very familiar with the measure. Second, the measure is internationally very well accepted and psychometrically sound [27]. Third, EDSS scores are statistically easy to handle. Fourth, to our knowledge, there is no reliable and easy-to-handle measure to both quickly and reliably assess the disease status of a person with neurological diseases such as NMOSD and MS.

## 2. Methods 

### 2.1. Data Source

We used data from the Isfahan Hakim MS database (IHMSD; Isfahan, Iran). This database contains demographic and clinical data of patients with MS, NMOSD, and related diseases who are routinely assessed as at the Outpatient MS Clinic of the Kashani Hospital (affiliated with the Isfahan University of Medical Sciences, Isfahan, Iran). The clinic is the major tertiary referral center and covers almost the whole population in Isfahan province (5.1 million in 2016). To diagnose NMOSD, the chief neurologist (VS), along with further male and female neurologists specifically trained for MS and CNS demyelinated disorders assessed all patients and diagnosed NMOSD based on the international consensus diagnostic criteria [28,29]. Clinical and para-clinical information were updated every six months (please see [30] for further details on the data set). The ethical committee of the Isfahan University of Medical Sciences (IUMS; Isfahan, Iran) approved the study (ethical code: IR.MUI.MED.REC.1400.268), which was performed in accordance with the rules laid down in the Declaration of Helsinki and its later amendments [31]. 

### 2.2. Study Population

In this monocentric and retrospective study, data of individuals with confirmed NMOSD were considered. Inclusion criteria were: 1. Age of at least 18 years at disease onset. 2. Diagnosis of NMOSD, based on the international consensus diagnostic criteria [28,29], and as ascertained by experienced and independent neurologists. 3. Signed written informed consent; patients signed a general written informed consent to agree that fully anonymized data in the database can be used for research. Exclusion criteria were: 1. insufficient data; 2. indefinite NMOSD diagnosis; 3. diagnosis of other demyelinating or similar disorders such as myelin oligodendrocyte glycoprotein antibody disorders (MOGAD), recurrent idiopathic optic neuritis (RION), chronic RION (CRION), or idiopathic transverse myelitis, as ascertained by experienced, independent and trained neurologists and based on a thorough clinical and lab assessment. 

### 2.3. Demographic and Clinical Data

We extracted the demographic and clinical information from the database. Demographic variables included age (years) and sex (male, female). Clinical variables were:

Age at initial symptoms: patients’ age at first symptoms or sign of NMOSD. 

First inter-attack interval: time interval between first and second attack. 

Initial symptoms: type of onset was categorized as visual symptom (ON), motor symptom, sensory symptom, brainstem syndrome, cerebral syndrome, and other.

NMOSD clinical symptoms during follow-up: ON, TM, area postrema syndrome, acute brainstem syndrome, and symptomatic cerebral syndrome. 

Expanded Disability Status Scale (EDSS): trained neurologists assessed individuals’ degree of disability with the EDSS [25]. The EDSS is an accepted and widely used tool for objective assessment of the disability levels of individuals with MS, CIS or NMOSD. The total score is on a scale from 0 to 10, with increments of 0.5–1.0, and with higher scores reflecting higher levels of disability. EDSS scores between 1.0–4.5 refer to people who are fully ambulatory. EDSS scores between 5.0–9.5 reflect an impairment to ambulation. Meyer-Moock et al. [27] reported in their systematic review the high validity and reliability of the EDSS. Meyer-Moock et al. [27] also concluded that the EDSS is suitable for describing the clinical status, the degree of physical disability, and for monitoring the disease progression.

Course of disease: the course of disease was defined as monophasic and relapsing. Relapsing means: patients experienced relapses and remissions after disease onset. Monophasic means: patients had no relapses between disease onset and the last assessment [32]. 

Comorbidities: the history of comorbidity between disease onset and the subsequent visits was described based on the chart review [33]. Comorbidities were categorized as autoimmune, somatic and psychological.

Family history of CNS demyelinating disorders: diagnosis of NMOSD, MS and other demyelinating diseases in first relative family members. 

Annualized relapse rate (ARR): ARR calculated as the total number of relapses experienced divided by duration of disease.

Progression index: PI was calculated as the last EDSS score divided by duration of disease. A higher ratio indicates a higher EDSS score or a shorter disease duration or both; a lower ratio indicates a lower EDSS score or a longer disease duration or both. 

Medication: medication refers to the medications used to manage NMOSD, such as interferon beta-1a, glatiramer acetate, rituximab, teriflunomide, azathioprine, prednisolone acetate or methotrexate. 

### 2.4. Para-Clinical Findings

All patients were tested for aquaporin-4-antibody (AQP4-Ab) by the cell-based assay method. Patients with NMOSD with negative AQP4-Ab, along with other patients suspected to be affected with NMOSD, were tested for anti-myelin oligodendrocyte glycoprotein (MOG) antibody using cell-based assay method. As stated above, patients with MOGAD were excluded from the study.

Our database also included findings of both brain and spinal magnetic resonance imaging (MRI) at first visit. Spinal MRI: detected type of lesion (presence or absence of spinal cord lesion and longitudinally extensive transverse myelitis [LETM]); location of lesion (cervical or thoracic, cervical, thoracic, or lumbar); number of lesions, and section of lesions (central, peripheral, or both). The findings of brain MRI were reported as normal or abnormal [34]. We report data from the spinal imaging taken at the last visit.

### 2.5. Statistical Analysis

For continuous variables such as age, progression index, or the total number of attacks at follow-up, a series of ANOVAs was performed with group (deteriorating; unchanged; improved) as independent factor. 

For categorical variables such as gender, first symptoms, course of disease, medication, a series of X2-tests was performed. We used Mann–Whitney U test to compare EDSS score at baseline, EDSS score at last visit, and EDSS change between seropositive and seronegative NMOSD patients.

The level of significance was set at alpha < 0.05, and all statistical computations were performed with SPSS^®^ 28 (IBM Corporation, Armonk, NY, USA) for Apple Mac^®^ (Cupertino, CA, USA). 

## 3. Results

Characteristics of patients are summarized in Table 2. Of all NMOSD patients who met the 2015 NMOSD criteria, 28 (48.3%) did not satisfy 2005 criteria. Of eight patients with monophasic course, two were seropositive. One of them experienced LETM and another one had ON and area postrema syndrome. Among six seronegative patients with monophasic course, four experienced ON and LETM, and each one had ON + area postrema syndrome and LETM + area postrema syndrome. 

Among 28 relapsing patients with negative AQP4-Ab, 19 presented ON and TM, 1 had ON + TM + area postrema syndrome, 3 patients experienced TM and area postrema syndrome, 1 patient presented TM + area postrema syndrome + acute brainstem syndrome, and 4 patients experienced ON and area postrema syndrome during follow-up. 

Among 22 relapsing patients with positive AQP4-Ab, 5 presented both ON and TM, 7 experienced only ON, 4 experienced only TM, 1 presented ON + TM + acute brainstem syndrome + symptomatic cerebral syndrome, 3 presented only area postrema syndrome, and one each developed acute brainstem syndrome and symptomatic cerebral syndrome. 

The outcome, and thus the independent variable, was the difference between the EDSS median at baseline and the EDSS median at follow-up. An increased score from baseline to follow-up was labelled “deteriorating”; an unchanged score from baseline to follow-up was labelled “unchanged”, and a decreased score from baseline to follow-up was labelled “improved”. Accordingly, based on their individual EDSS score differences, participants were assigned to those who deteriorated, those who remained unchanged, and those who improved. Results are reported in Table 3.

Table 2 provides the descriptive and inferential statistical overview of all variables, separated between those patients who deteriorated (*n* = 6), those who remained unchanged (*n* = 13), and those who improved (*n* = 39) from disease onset to the follow-up about five years later.

EDSS scores did not statistically or descriptively differ at baseline between the three groups. 

As regards continuous variables, compared to those who improved, patients who deteriorated had a statistically significantly higher number of attacks at follow-up, and a higher progression index, though the annualized relapse rates at follow-up did not statistically differ. Further, for age, age at follow-up, and BMI, regarding the interval between the first and second attack, no statistically or descriptively significant differences between the three groups were observed. 

As regards the categorical variables, those patients who did not change were descriptively more seronegative than statistically expected and less seropositive than statistically expected. For those participants who improved, descriptively less were seronegative than statistically expected, and more were seropositive than statistically expected. Further, for those who improved, more than statistically expected were treated with rituximab. 

Next, those who deteriorated had descriptively more active cervical plaques than statistically expected, though the number of cases was modest to make a precise statement. 

Next, for the following categories, no statistic or descriptive differences were observed: family history of CNS demyelinating disorders, somatic, autoimmune and psychiatric comorbidities at follow-up, sleep problems, structural neurological indices such as the occurrence and localization of plaques, or longitudinally extensive transverse myelitis (LETM). 

A key factor to define NMOSD are visual impairments. We asked if the occurrence of visual impairments at baseline statistically significantly differed between the three groups, and the answer was no. Table 4 shows the descriptive and statistical indices of visual impairments (yes vs. no). The X^2^-test was not significant, or simply put: among those participants, who did deteriorate, or not change, or improve over time, visual impairments at baseline were evenly distributed. 

## 4. Discussion

Among a smaller, though thoroughly examined sample of 58 patients with NMOSD reassessed about five years after disease onset, the key findings of the present retrospective and observational study were as follows: First, three prototypical courses could be identified. Such disease courses were based on the change of EDSS scores from baseline at disease onset to the follow-up about five years later. Of the 58 patients assessed, 6 did deteriorate, 13 remained unchanged, and 39 improved. Second, deterioration was associated with a higher total number of attacks, and third with a higher progression index, when compared to those patients who did improve. Fourth, the status of deterioration, no change and improvement were not associated with the annualized relapse rate (ARR) or with the EDSS scores at baseline. Fifth, data from brain and spinal cord MRIs appeared to have no predictive value. Sixth, improvements were associated with the administration of rituximab. Seventh, improvements were associated with a lower rate of seronegative cases, and with a higher rate of seropositive cases.

Given this, we claim that the present pattern can expand upon to the current literature in the following four ways: First, deterioration was associated with a higher total number of attacks, and second, with a higher progression index. Third, the type of NMOSD (seronegative; seropositive) was descriptively associated with no change or with improvements. Fourth, rituximab was more observed among those who improved. Rituximab is a B-cell-targeting and chimeric anti-CD20 monoclonal antibody (mAb). As such, it is not considered a cytotoxic agent.

Here, we descriptively compare previous findings with the current data.

Aungsumart and Apiwattanakul [16] identified a minimal disability such as preserved reflexes at disease onset as a favorable predictor for the outcome of plasma exchange treat NMOSD. Here, based on the EDSS scores at baseline, we were unable to identify the disability status as predictor for the course of illness. 

Carnero Contentti, Hryb, Morales, Gomez, Chiganer, Di Pace, Lessa and Perassolo [18] were unable to identify robust factors to predict the illness course among 21 individuals with MNOSD and LETM. In contrast, in the present study, we could observe that a higher number of attacks and a higher progression index was associated with an unfavorable disease progression. 

Bălaşa, Maier, Bajko, Motataianu, Crişan and Bălaşa [19] identified more motor and visual impairments as specific predictors, while in the present study, the overall EDSS score at follow-up, but not at disease onset, was indicative for the illness course. Bălaşa, Maier, Bajko, Motataianu, Crişan and Bălaşa [19] also further observed indices of CSF and spinal MRI, and the administration of immunosuppressive medications to favorably impact the illness course. Similarly, in the present study, the administration of rituximab was associated with the illness improvements. 

As regards the annual relapse rate (ARR), an average ratio of 0.5 (deteriorating: 0.71; unchanged: 0.47; improving: 0.41) was descriptively lower compared to the ratio of 0.8 recently reported from a larger retrospective and observational cohort study [21] and compared to the ratio of 0.6 among 144 patients with NMOSD followed-up 1.2 years after disease onset [22].

The novelty of the results should be balanced against the following limitations. First, the sample size may be considered small, though this also allowed a thorough and deepened data analysis. Second, only full data sets were considered to avoid an error-prone data analysis, which almost by nature might lead to biased patterns of results. Third, the quality of the data did not allow a thorough analysis of the medication treatment; this holds particularly true as regards possible switches of medications over time. Fourth, to distinguish participants who deteriorated, improved or remained stable over a time lapse of about five years, we calculated the differences between the overall EDSS scores at baseline and at follow-up. We are aware that this algorithm bears the risk to overlook specific nuances of disease progress. To illustrate, a person might deteriorate in their walking ability, while vision improved; in contrast, another person might deteriorate in their vision, while walking ability improved. Following this, the EDSS overall score remained unchanged, though the two persons might report completely different levels of quality of life. Fifth, it is conceivable that latent and unassessed factors might have biased two or more dimensions in the same or opposite directions. Sixth, two major issues must be considered: First, the lack of CSF findings might bear the risk of the blurred exclusion of alternative diagnosis as a key element of the NMOSD diagnostic criteria [1,2,3,4,5,6,7]. Relatedly, seventh, at the second time point, visual acuity was not assessed. However, as shown in Table 4, the distribution of visual impairments at baseline did not statistically differ between the three groups. As such, it might be conceivable, though not provable with the current data, that the occurrence of visual impairments might have been similar also at the second timepoint. To counter these two limitations, future studies focusing on the long-term progress of individuals with NMOSD should consider both CSF findings and the thorough assessment of their visual acuity.

## 5. Conclusions

Among a smaller sample of individuals with NMOSD three prototypical illness courses could be identified. The mean time lapse from disease onset to follow-up was five years. A total of 39 out of 58 individuals did improve, while 13 out of 58 did neither improve nor deteriorate; by contrast, 6 out of 58 deteriorated, or simply put, while 10.4% the illness course was unfavorable, this was not the case for 89.6%. Deterioration was associated with a higher number of attacks and a higher progression index. Overall, despite the prototypical illness course, differences between the three groups as regards illness-related and sociodemographic dimensions were modest, suggesting that a highly individualized and tailored treatment regimen appears mandatory. 

## Figures and Tables

**Table 1 jcm-11-00734-t001:** Comparison of sociodemographic and clinical characteristics between Neuromyelitis Optica Spectrum Disorder (NMOSD), Multiple Sclerosis (MS), and Myelin Glycoprotein Antibody Disorders (MOGAD).

	Neuromyelitis Optica Spectrum Disorder (NMOSD)	Multiple Sclerosis (MS)	Myelin Glycoprotein Antibody Disorders (MOGAD)
Prevalence	Worldwide prevalence: 0.7–10 per 100,000 persons Prevalence in Iran: 0.86–1.9 per 100,000 persons	Worldwide prevalence: 35.9 per 100,000 persons Prevalence in Iran: 29.3 per 100,000 persons	Worldwide prevalence: 2.0 per 100,000 persons Prevalence in Iran is unknown
Mean age at disease onset	40	30	Most in children
Female to male ratio	2–9:1	2–4:1	Around 1:1
Serology	AQP4-Ab	No specific antibody	MOG-IgG
Disease course	Relapsing and rarely monophasic	RRMS, SPMS, PPMS	Relapsing and monophasic
Main clinical features	ON, LETM, postrema syndrome	Optic neuritis Myelitis Brain syndromes	Optic neuritis Myelitis ADEM/MDEM Brainstem/cerebral cortical encephalitis Cranial nerve involvement
Optic neuritis	Mostly bilateral	Mostly unilateral	Mostly bilateral
Myelitis	Long	Short	Often long
Attack disability	High	Low	High
Attack recovery	Poor	Fair to good	Fair to good
Autoimmune comorbidity	Common	Rare	Rare
Treatment	IST, some DMTs such as interferon and fingolimod ae harmful	DMTs	IST if recurrent

Notes: RRMS = relapsing remitting multiple sclerosis; SPMS = secondary progressive multiple sclerosis; PPMS = primary progressive multiple sclerosis; ON = optic neuritis; LETM = longitudinal transverse myelitis; ADEM = Acute Disseminated Encephalomyelitis; MDEM = Multiphasic Disseminated Encephalomyelitis.

**Table 2 jcm-11-00734-t002:** Sociodemographic and illness-related information between seropositive and seronegative individuals with MNOSD.

Dimensions		Group	Statistics
	Seropositive	Seronegative	
N	24	34	
	Mean (SD)	Mean (SD)	
Age at disease onset (years)	30.87 (8.89)	29.70 (81.52)	t(56) = 0.516
Age at follow-up	35.75 (9.74)	33.59 (9.78)	t (56) = 0.832
Age difference between disease onset and follow-up (years) = disease duration	5.20 (4.80)	3.94 (4.39)	t(56) = 1.040
BMI at follow-up	23.82 (4.04)	26.29 (5.42)	t(56) = −1.886
Total number of attacks at follow-up			
Annualized relapse rate at follow-up (ARR)	0.39 (0.36)	0.48 (0.54)	t(56) = −0.716
Interval (years) between first and second attack	2.40 (2.54)	2.35 (2.035)	t(25) = 0.053
Progression Index (last EDSS score: disease duration)			t(45) = −0.831
	n/n	n/n	X^2^-tests
Gender (female/male)	20/4	30/4	X^2^(*N* = 58, df = 1) = 0.284
First symptoms (visual, sensory, motor, brainstem, cerebellar, other)	12/5/3/1/1/2	21/6/6/0/0/1	X^2^(*N* = 58, df = 5) = 4.282
Course of disease (monophasic; relapsing)	22/2	28/6	X^2^(*N* = 58, df = 1) = 1.026
Medications (rituximab/ azathioprine/prednisolone acetate/methotrexate)	12/11/0/3	29/4/1/1	X^2^(*N* = 44, df = 3) = 4.875
Family history of multiple sclerosis (yes/no)	5/19	8/26	X^2^(*N* = 58, df = 1) = 0.059
Family history of CNS demyelinating disorders (yes/no)	2/22	2/32	X^2^(*N* = 58, df = 1) = 0.132
Somatic comorbidities at follow-up (no/yes)	12/12	25/9	X^2^(*N* = 58, df = 1) = 3.372
Psychiatric comorbidities at follow-up (yes/no)	4/20	11/23	X^2^(*N* = 58, df = 1) = 1.806
Autoimmune comorbidities at follow-up (yes/no)	1/23	1/33	X^2^(*N* = 58, df = 1) = 0.063
History of subjective sleep problems (yes/no)	16/8	24/10	X^2^(*N* = 58, df = 1) = 0.101
Location of brain plaques at baseline (supratentorial/ infratentorial/whole brain)	5/1/6	14/0/9	X^2^(*N* = 35, df = 2) = 2.670
Active plaque in the brain at baseline (yes/no)	1/23	2/30	X^2^(*N* = 56, df = 1) = 0.117
Number of active plaques in the brain at baseline (1/3/10)	1/0/0	1/1/0	X^2^(*N* = 3, df = 1) = 3.000
Active cervical plaque at baseline (yes/no)	2/22	0/29	X^2^(*N* = 51, df = 1) = 2.744
Localization of cervical plaques at baseline (central/peripheral/central and peripheral)	5/0/10	10/2/16	X^2^(*N* = 43, df = 2) = 1.234
LETM at baseline	12/12	16/18	X^2^(*N* = 58, df = 1) = 0.049
LETM at last visit (present/absence)	12/12	29/5	X^2^(*N* = 58, df = 1) = 8.458 **
	Median (IQR)	Median (IQR)	
EDSS score baseline	2.5 (1)	2.0 (0.625)	U = 301.50
EDSS score follow-up	1.25 (2.375)	1.5 (2)	U = 399.50
EDSS score; differences	1.0 (0)	1.0 (1.0)	U = 282.0 *

Notes: BMI = body mass index; EDSS = Expanded Disease Status Scale; * = *p* < 0.05, ** = *p* < 0.01.

**Table 3 jcm-11-00734-t003:** Overview of descriptive and inferential statistical indices of sociodemographic, disease-related and treatment-related information, separately for the three groups of improving (*n* = 39), unchanged (*n* = 13), and deteriorating (*n* = 6).

Dimensions		Group		Statistics
	Deteriorated	Unchanged	Improved	
N	6	13	39	
	M (SD)	M (SD)	M (SD)	
Age at disease onset (years)	34.0 (9.34)	29.76 (6.15)	29.74 (8.95)	F(2, 55) = 0.68, η_p_^2^ = 0.024 (S)
Age at follow-up	41.50 (11.32)	32.54 (6.05)	34.05 (10.23)	F(2, 55) = 1.91, η_p_^2^ = 0.065 (M)
Age difference between disease onset and follow-up (years) = disease duration	7.50 (6.57)	3.38 (3.55)	4.36 (4.46)	F(2, 55) = 1.74, η_p_^2^ = 0.059 (S)
BMI at follow-up	25.33 (5.30)	25.42 (4.03)	25.22 (5.37)	F(2, 55) = 0.00, η_p_^2^ = 0.00 (T)
Total number of attacks at follow-up	2.67 (1.75)	1.23 (0.60)	1.28 (1.07)	F(2, 55) = 4.56 *, η_p_^2^ = 0.142 (L) Deteriorated > unchanged
Annualized relapse rate at follow-up (ARR)	0.62 (0.54)	0.47 (0.60	0.41 (0.43)	F(2, 55) = 0.48, η_p_^2^ = 0.017 (T)
Interval (years) between first and second attack	3.33 (2.51)	2.33 (3.01)	2.22 (1.93)	F(2, 24) = 0.36, η_p_^2^ = 0.019 (T)
Progression Index (last EDSS score: disease duration)	0.71 (0.29)	0.75 (0.95)	0.25 (0.35)	F(2, 44) = 4.48 *, η_p_^2^ = 0.169 (L) Deteriorated > improved
	n/n	n/n	n/n	X^2^-tests
Gender (female/male)	6/0	12/1	32/7	X^2^(*N* = 58, df = 2) = 1.93
First symptoms (visual, sensory, motor, brainstem, cerebellar, other)	5/0/1/0/0/0	9/1/1/1/0/1	19/10/7/0/1/2	X^2^(*N* = 58, df = 10) = 9.55
Course of disease (monophasic; relapsing)	0/6	1/12	7/32	X^2^(*N* = 58, df = 2) = 1.93
NMOSD-type (seronegative/seropositive)	3/3	11/2 Negative: more than statistically expected; positive: less than statistically expected	20/19 Negative: less than statistically expected; positive: more than statistically expected	X^2^(*N* = 58, df = 2) = 4.67 ^(^*^)^
Medications (rituximab/ azathioprine/prednisolone acetate/methotrexate)	2/3/0/0	11/1/0/0	28/11/1/4 Rituximab; more than statistically expected	X^2^(*N* = 44, df = 6) = 23.868 **
Family history of multiple sclerosis (yes/no)	1/5	4/9	8/31	X^2^(*N* = 58, df = 2) = 0.72
Family history of CNS demyelinating disorders (yes/no)	0/6	0/13	4/13	X^2^(*N* = 58, df = 2) = 2.09
Somatic comorbidities at follow-up (no/yes)	3/3	4/9	14/25	X^2^(*N* = 58, df = 2) = 0.66
Psychiatric comorbidities at follow-up (yes/no)	2/4	6/7	7/32	X^2^(*N* = 58, df = 2) = 4.24
Autoimmune comorbidities at follow-up (yes/no)	0/6	0/13	2/37	X^2^(*N* = 58, df = 2) = 1.00
History of subjective sleep-problems (yes/no)	4/2	10/3	26/13	X^2^(*N* = 58, df = 2) = 0.50
Location of brain plaques at baseline (supratentorial/ infratentorial/whole brain)	1/0/2	5/0/5	13/1/8	X^2^(*N* = 35, df = 4) = 1.69
Active plaque in the brain at baseline (yes/no)	0/5	2/10	1/38	X^2^(*N* = 56, df = 2) = 3.91
Number of active plaques in the brain at baseline (1/3/10)	-	1/1/0	0/0/1	X^2^(*N* = 3, df = 2) = 3.00
Active cervical plaque at baseline (yes/no)	1/0 Yes: more than statistically expected; No: less than statistically expected	0/11	1/35	X^2^(*N* = 51, df = 2) = 5.29 ^(^*^)^
Localization of cervical plaques at baseline (central/peripheral/central and peripheral)	1/0/2	5/1/4	9/1/20	X^2^(*N* = 43, df = 4) = 2.63
Brain atrophy at baseline (no/mild/moderate)	5/0/0	11/1/0	37/0/2	X^2^(*N* = 56, df = 4) = 4.57
Longitudinally extensive transverse myelitis at last visit (yes/no)	3/1	12/6	26/10	X^2^(*N* = 58, df = 2) = 0.217
	Median (range)	Median (range)	Median (range)	
EDSS score baseline	2 (4)	2 (3)	2.5 (6)	F(2, 55) = 1.09, η_p_^2^ = 0.038 (S)
EDSS score follow-up	3 (4)	2 (3)	1 (6)	F(2, 55) = 16.39 ***, η_p_^2^ = 0.373 (L)
EDSS score; differences	1 (2)	0 (0)	−1.5 (2)	F(2, 55) = 85.64 ***, η_p_^2^ = 0.757 (L)

Notes: BMI = body mass index; EDSS = Expanded Disease Status Scale; ≥ statistically significantly higher; * = *p* < 0.10; * = *p* < 0.05, ** = *p* < 0.01, *** = *p* < 0.001. (T) = trivial effect size; (S) = small effect size; (M) = medium effect size; (L) = large effect size.

**Table 4 jcm-11-00734-t004:** Distribution of visual impairments at baseline, separately for the three groups of improving (*n* = 39), unchanged (*n* = 13), and deteriorating (*n* = 6).

	Group						Statistics
	Deteriorated		Unchanged		Improved		
N	9		13		39		
	Yes	No	Yes	No	Yes	No	
Visual impairment at baseline	5	1	9	4	19	20	X^2^(*N* = 58; df = 2) = 3.58, *p* = 0.17

## Data Availability

Data are made available upon request to experts in the field and upon thorough explanations of why and how data are used.

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
