# Peer review of "Sociodemographic and Illness-Related Indicators to Predict the Status of Neuromyelitis Optica Spectrum Disorder (NMOSD) Five Years after Disease Onset"

_jcm, 2022, doi:10.3390/jcm11030734_

Round 1
Reviewer 1 Report
Thank you for submitting the revision. The authors have attempted to revise the manuscript based on previous reviews.
There are several methodical issues here:
- EDSS worsening is considered as the criteria for dividing the patients into three groups, and a measure which utilises EDSS (progression index) is then shown as a factor which is associated with worsening, which is a circular argument. (This may to some extent be relevant to the other "significant" variable - which is higher number of attacks).
- Even though the authors argue the benefit of EDSS as a criteria, this is not sensitive to pick up patients with predominant visual symptoms - and as per Table 3, 33 out of 61 patients had visual symptoms at onset which makes this scale inappropriate.
- The diagnosis in a good proportion of issues is seronegative and it would have been good to have CSF examination and also details of how other diagnoses (esp. MS) was excluded - since "exclusion of alternative diagnosis" is a key element of the 2015 NMOSD diagnostic criteria. I am not sure citing the lack of CSF findings as a limitation will be sufficient to justify this serious flaw.
- Similar to point 2 above, I am not entirely sure we can classify NMOSD as improved or unchanged without having visual acuity data.
Author Response
We thank Reviewer #1 for the care devoted to the present manuscript. The comments and suggestions helped us again to improve the quality of the manuscript. Please find the detailed point-by-point-response attached as a separate file.
Again, thank you very much for all your kind efforts.

Reviewer 2 Report
The authors made all recommended corrections and amendments.
Author Response
We thank Reviewer #2 for the care devoted to the present manuscript. The comments and suggestions helped us to improve the quality of the manuscript. Please find the detailed point-by-point-response attached as a separate file.
Again, thank you very much for all your kind efforts.

This manuscript is a resubmission of an earlier submission. The following is a list of the peer review reports and author responses from that submission.
Round 1
Reviewer 1 Report
Huge, valuable work.
Introduction: What are the known causes of NMOSD? Please consider to construct a table about the differential diagnosis of NMOSD! (MS, MOGAD, relationship between NMOSD and LETM, RION etc)
L189 Why was ANOVA preferred to non parametric tests? Was the normality checqued?
L190-191 Please describe the definition of the groups! What were the criteria of deteriorating, unchanged or improved status, respectively?
Author Response
We thank Reviewer #1 for the care devoted to the present manuscript. The comments and suggestions helped us to improve the quality of the manuscript. Please find the detailed point-by-point-response attached as a separate file.
Again, thank you very much for all your kind efforts.

Reviewer 2 Report
The authors discuss a very valid study trying to predict prognostic factors in NMOSD. The study includes 58 patients, of which 24 were positive for Aqp-4 antibodies. It would be good to see the break up of the 34 seronegative patients to see how they satisfy the recent diagnostic criteria. It might also be worthwhile dividing the group into seropositive and seronegative groups to ensure there is a homogenous comparison, although the numbers would become very small.
The main difficulty I have is in the definition of unchanged vs improved vs deteriorated. What is the definition for this? If there is a change by 0.5 or 1 score because of spasticity, is this considered as worsening. Why did the authors choose the EDSS at presentation, rather than at discharge following initial admission which would have given a better idea of the long-term prognosis. This is more likely to show the influence of future relapses and subsequently the effect of disease modifying therapy.
In my view, it would be better to look at the 5-year endpoint and divide patients into three groups based on their "final" EDSS scores and then retrospectively analyse whether the initial admission EDSS, acute treatments, initial symptoms or disease modifying treatments influenced the outcome.
How many of these patients had optic neuritis alone (I know 33 had visual symptoms), and EDSS may be a very poor marker to assess "improvement" in isolated optic neuritis patients.
13 patients had a family history of MS - which seems very high for a cohort of NMOSD patients - hence makes it even more important to find out how the diagnosis was confirmed in the large seronegative group. Was the data independently reviewed by a clinician not involved in the study, or was the "senior author's" opinion considered as the final opinion. I have seen several patients with optic neuritis and brain stem syndrome being diagnosed as NMOSD based on the current criteria, and many of them eventually turned out to have MS between 2 and 15 years later.
30 patients did not have LETM which seems a very high number, especially considering that nearly 60% patients were seronegative. Could this have influenced the conclusion.
I might have missed this - but how many had CSF and what proportion of these patients had oligoclonal bands? This is a big factor, especially with such a high proportion of seronegative patients.
In summary, even though there are reasonable number of patients described, we need to be absolutely confident that the included patients are clearly a homogenous group and also have well defined end points, without which any reasonable conclusion is tricky.
Author Response
We thank Reviewer #2 for the care devoted to the present manuscript. The comments and suggestions helped us to improve the quality of the manuscript. Please find the detailed point-by-point-response attached as a separate file.
Again, thank you very much for all your kind efforts.
